# Zonulin as a Potential Therapeutic Target in Microbiota-Gut-Brain Axis Disorders: Encouraging Results and Emerging Questions

**DOI:** 10.3390/ijms24087548

**Published:** 2023-04-19

**Authors:** Apor Veres-Székely, Csenge Szász, Domonkos Pap, Beáta Szebeni, Péter Bokrossy, Ádám Vannay

**Affiliations:** 1Pediatric Center, MTA Center of Excellence, Semmelweis University, 1083 Budapest, Hungary; veres-szekely.apor@med.semmelweis-univ.hu (A.V.-S.);; 2ELKH-SE Pediatrics and Nephrology Research Group, 1052 Budapest, Hungary

**Keywords:** zonulin, zonula occludens 1, microbiota, gut, brain, dysbiosis, larazotide acetate, permeability, tight junction, barrier

## Abstract

The relationship between dysbiosis and central nervous diseases has been proved in the last 10 years. Microbial alterations cause increased intestinal permeability, and the penetration of bacterial fragment and toxins induces local and systemic inflammatory processes, affecting distant organs, including the brain. Therefore, the integrity of the intestinal epithelial barrier plays a central role in the microbiota–gut–brain axis. In this review, we discuss recent findings on zonulin, an important tight junction regulator of intestinal epithelial cells, which is assumed to play a key role in maintaining of the blood–brain barrier function. In addition to focusing on the effect of microbiome on intestinal zonulin release, we also summarize potential pharmaceutical approaches to modulate zonulin-associated pathways with larazotide acetate and other zonulin receptor agonists or antagonists. The present review also addresses the emerging issues, including the use of misleading nomenclature or the unsolved questions about the exact protein sequence of zonulin.

## 1. Introduction

Emerging literary data have revealed a dynamic bidirectional interaction between gut microbiota and the nervous system, described by the microbiota–gut–brain axis (MGBA) [1].

Various factors regulate microbial composition, including stress, nutritional habits, environmental impacts, and in parallel, luminal agents in gut affect neural functions [1,2]. The complex communication between microbiota and brain is continuous via inflammatory mediators, neurotransmitters, neuroactive microbial metabolites, and vagus and enteric nerves, among others [1]. The importance of MGBA was revealed by numerous human studies that demonstrated a correlation between altered composition of gut microbiota and neurological disorders, including Parkinson’s disease, Alzheimer’s disease, anxiety, depression, and autism both in children and adults [2,3,4]. Intestinal permeability is a key factor in this process determining the penetration of luminal elements into the circulation [5]. The barrier function of the intestinal epithelial layer is provided by intercellular junctions, including tight junction (TJ), adherens junction, and desmosome [6]. In the development of “leaky gut”, TJs and their component, the zonula occludens 1 (ZO-1), play a crucial role. ZO-1, also known as tight junction protein 1 (TJP-1), is a membrane-associated protein that ensures the basolateral cell–cell adherence of intestinal epithelial cells by cross-linking the TJ transmembrane proteins (claudin, occludin, junction adhesion molecule) and the actin cytoskeleton [7,8].

In 2000, the research group of Fasano reported the discovery of zonulin, a human protein analogue of the *Vibrio cholerae*-derived Zonula occludens toxin (Zot), regulating paracellular permeability through protein kinase C (PKC)-dependent rearrangement of actin microfilaments and deterioration of ZO-1 structure [9]. Since then, the effect of the zonulin pathway on the regulation of intestinal permeability has been supported by phase 2 clinical studies demonstrating the beneficial effect of the zonulin antagonist larazotide acetate (AT-1001) in patients with celiac disease [10]. Although the most of our knowledge about zonulin is related to intestinal diseases, its importance in almost all our organs, including brain, heart, lung, kidney, liver, skin, etc., has now been described [11,12,13,14,15,16,17]. Indeed, dysbiosis is associated with increased intestinal zonulin release, impaired gut permeability, and upregulation of inflammatory mediators. The spread of gut-derived microbial fragments, toxins, and inflammatory factors, including zonulin, finally reach distant organs, including the central nervous system, leading to increased blood–brain barrier (BBB) permeability, neuroinflammation, and behavioral changes that are partially ameliorated by microbiota depletion [18]. All these together suggest that dysbiosis and the zonulin pathway may be central factors in MGBA-related diseases.

In this review, we aimed to summarize the latest results about the zonulin pathway, focusing on the regulatory effect of microbiota on zonulin release, the relationship between zonulin and the central nervous system (CNS), and the possible zonulin-related therapeutic opportunities targeting TJs, in particular ZO-1. Due to the similar appellations of ZO-1, Zot, and zonulin, these molecules are often confused in the literature. Therefore, as a part of this article, we try to draw attention to the misunderstandings arising from imprecise wording, as well the known technical issues and limitations of zonulin-related research. Our review processes a wide variety of literary data, including clinical observations, clinical trials, in vitro, and in vivo experimental data; therefore, we hope that it will prove useful to those involved in translational research on zonulin and MGBA.

## 2. Zonulin

### 2.1. Zonulin as Pre-Haptoglobin 2

In the 1990s, a novel toxin secreted by *Vibrio cholera*, called Zot or zonula occludens toxin, was described. Zot interacts with its specific cell surface receptor present in the gut [19] and brain [20] and induces PKC-dependent polymerization of actin microfilaments thereby regulating TJs and increasing the permeability of the epithelial layer [21,22,23,24]. In addition to *Vibrio cholerae*, other *Vibrio* strains can produce similar 3D structure proteins, causing cytoskeletal disruption of epithelial cells [25]. Moreover, *Campylobacter* spp., including *Campylobacter concisus,* can also release Zot, and although it has only 16% amino acid identity compared to *Vibrio cholerae* Zot, it still induces intestinal epithelial barrier damage [26,27,28].

In 2000, Wang et al. reported a protein isolated from the human intestine, sharing significant structural and biological similarities with Zot, derived from *Vibrio cholerae*, it was therefore named zonulin [11]. In this study, zonulin was purified from mucosal lysates using anti-Zot antibody affinity columns, and it was demonstrated that the exposure of intestinal tissue to zonulin decreases the transepithelial resistance in an Ussing chamber. In the same year, Fasano et al. published their findings on the elevated level of zonulin in the intestinal tissue of patients with active celiac disease [9].

Later, Tripathi et al. demonstrated that zonulin is identical to pre-haptoglobin 2, an inactive precursor of haptoglobin 2 [7]. Haptoglobins are secretory proteins, belonging to the acute-phase plasma proteins [29]. Although a large amount of haptoglobin is present in the serum under physiological conditions, its production is upregulated by major inflammatory cytokines, including IL-1, IL-6, and TNF-α [29,30,31]. The primary function of haptoglobins is to eliminate the hemoglobin released from lysed red blood cells, which could cause tissue damage due to its strong oxidative and proinflammatory effect [32]. Haptoglobins form a stable covalent bond with hemoglobin, thereby stabilizing it in a reduced state and facilitating its binding to CD163 receptor expressed on macrophages, thereby accelerating the clearance of hemoglobin via endocytosis [29].

Haptoglobin has two genetic variants, haptoglobin 1 and 2, resulting in three possible phenotypes (1-1 homozygote, 2-1 heterozygote, and 2-2 homozygote) in humans [33]. Pre-haptoglobin 2 is the primary translation product of the haptoglobin 2 mRNA, found in individuals with heterozygous or homozygous haptoglobin 2 genotypes [7,34,35]. The pre-haptoglobin 2 goes through a complex maturation process to reach its active form, including proteolytic cleavage in the endoplasmic reticulum, formation of disulphide bonds, dimerization, and other post-translational modifications, such as glycosylation, acetylation, iodination, or nitration [31,33]. In this process, the cleavage enzyme protease complement C1r subcomponent-like protein (C1r-LP) plays a crucial role [36,37,38], and thus is also hypothesized by Fasano to modulate the amount of zonulin in the circulation [39]. Serum level of pre-haptoglobin 2 or zonulin is approximately one thousandth of mature haptoglobins [7] and do not form complexes with hemoglobin [36,40].

### 2.2. Regulation of Zonulin

The liver is known as the major source of haptoglobins and C1r-LP; however, intestinal mucosal biopsies, organoids, and epithelial cell cultures have shown that large amounts of zonulin can also be released from the intestine [9,11,41,42].

The main regulator of zonulin release in the gut is C-X-C chemokine receptor type 3 (CXCR3) [42,43], which is an inflammatory chemokine receptor, characterized by a versatile ligand profile, including members of the interferon-γ-induced C-X-C motif chemokine ligand (CXCL) family. The primary role of CXCR3 is to induce chemotaxis, cell migration, and adhesion of immune cells [44,45]. Recently, CXCR3 has also been shown to be present in the intestinal lamina propria and epithelial cells, and its expression is upregulated in the inflamed intestine of patients with celiac or inflammatory bowel diseases [43,46,47]. In addition, luminal agents, including microbial and nutritional components (which represent important members of the MGBA) can also activate CXCR3-dependent zonulin release [48]. Indeed, using CXCR3 knock-out mice and various ex vivo and in vitro models, Lammers et al. demonstrated that CXCR3 activation by gliadin fragments led to myeloid differentiation primary response 88 (MyD88)-dependent zonulin release from intestinal epithelial cells [43,49]. MyD88 is an intracellular adaptor molecule for cell surface receptors such as Toll-like receptors (TLRs) and interleukin 1 receptors, and its primary role is to induce transcription by nuclear translocation of transcription factors, including interleukin regulatory factor (IRF) proteins and nuclear factor-κB (NF-κB) [50]. Although the gliadin-induced zonulin release has been found to be associated with celiac disease, the harmful effects of gluten exposition on intestinal epithelial cell viability and permeability have also been described in non-celiac patients [6,51]. Therefore, understanding the complex role of zonulin may also contribute to the development of therapy against other diseases. Disorders associated with abnormal zonulin levels will be discussed later in Section 3. Zonulin-related diseases.

Recently, the pivotal impact of microbiota on zonulin release has also been described; however, the underlying mechanism is partly unclear. Several studies have shown that bacterial lipopolysaccharide (LPS), derived from *Escherichia coli*, induces zonulin release in CaCo2 colon epithelial cells [52], whereas Thomas et al. found no effect on macrophages [49]. In addition, work by others has shown that treatment with LPS can also increase the expression of CXCR3 in epithelial and endothelial cells in vitro and in vivo [53,54]. Zhang et al. proved on CXCR3 knock-out mice that LPS-induced intestinal dysfunction and barrier damage is a CXCR3-dependent mechanism related to the NF-κB signalling pathway [55]. Lauxmann et al. drew attention to the structural similarity between gliadin fragments and certain parasite proteins and showed that the polyQ sequences of coccidian proteins can bind to the intestinal CXCR3 receptor, leading to an increase in intestinal permeability, thereby promoting parasite invasion into the lamina propria [56]. Indeed, the possible connection between zonulin and parasitic infections, such as malaria, was suggested by genetic studies, demonstrating an increased allele frequency in disease population [57]. Moreover, another study reported that elevated fecal zonulin levels were associated with fungal and parasitic overgrowth in stool samples [58]. These preliminary findings suggest that zonulin may play a role in parasitic infections.

Nevertheless, the link between gut microbiota and the regulation of zonulin release is unquestionable. Numerous studies have aimed to explore the effect of various species of bacteria (without expressing Zot) on zonulin levels in both descriptive (Table 1) and interventional (Table 2) human studies.

Related findings from experimental studies on cell lines and animal models are summarized in Table 3. Briefly, several Gram-negative bacterial strains, including *Escherichia coli*, *Prevotella*, *Pseudomonas*, and *Salmonella* spp., induce intestinal zonulin release, whereas others, mostly Gram-positive strains, such as *Bifidobacterium* and *Lactobacillus* spp., decrease zonulin levels (figure in Section 6). A possible mechanism underlying the protective effects of *Bifidobacterium* and *Lactobacillus* spp. is that these bacteria can cleave gluten peptides via hydrolyzing enzymes, thereby inhibiting the gliadin-induced cytotoxic responses in intestinal epithelial cells [87,88,89]. The presence of an additional pathway is suggested by recent findings demonstrating that heat-killed *Bifidobacterium* [90] and *Lactobacillus* [91] spp. (which do not produce active enzymes) still have beneficial effects on epithelial barrier function.

However, the therapeutic applicability of protective bacteria was proved in a few clinical studies, although further research is needed to reveal the most promising bacterial strains and to develop effective medical products containing the optimal mixture of probiotics and prebiotics.

### 2.3. Biological Activity of Zonulin

Tripathi et al. found that zonulin contains an epidermal growth factor (EGF)-like and also a proteinase-activated receptor 2 (PAR_2_) activating peptide-like motif, both necessary to activate EGFR [7]. Indeed, zonulin has been shown to fail to induce EGFR phosphorylation in PAR_2_ knock-down cells or knock-out mice, suggesting the importance of PAR_2_-induced transactivation of EGFR.

Since then, several studies have shown that crosstalk between EGFR and PAR_2_ via Ras-MAP-kinase pathway has a major impact on epithelial processes [109,110,111]. During its maturation, proteolytically cleaved zonulin (pre-haptoglobin 2) loses its EGFR activating capacity and does not increase intestinal permeability; however, it gains a new property for hemoglobin binding (see above in Section 2.1. Zonulin as pre-haptoglobin 2) [7].

Besides the transactivation of EGFR, PAR_2_-activation induces phosphatidyl inositol (PPI) turnover and stimulation of phospholipase C leading to diacylglycerol (DAG) activation and intracellular Ca^2+^ release through inositol 1,4,5-triphosphate (IP-3) increment, both of which induce PKC activation [112,113,114]. PKC activation causes depolymerisation and reduced peripheral density of the actin fibers, leading to cytoskeletal rearrangement and phosphorylation of ZO-1, causing its dislocation from the cell membrane [23,115,116,117]. As ZO-1 and actin fibers have a pivotal role in the maintenance of cell–cell adhesion of epithelial and endothelial cells, these processes lead to transient disassembly of the tight-junction complex and thus an increase in paracellular permeability [7,8,115,118,119] (figure in Section 6).

Interestingly, whereas zonulin-independent activation of PAR_2_ resulted in zonulin-like effects on the epithelial layer, leading to a decrease in transepithelial resistance (TER) and ZO-1 translocation [120,121,122], activation of EGFR by recombinant EGF had a protective effect on paracellular permeability and TJ integrity [123,124]. However, PAR_2_-mediated EGFR activation by house dust mite leads to decreased resistance and TJ disruption in bronchial epithelial cells [111]. Similarly, contradictory results have been obtained from studies using PAR_2_ or EGFR modulator compounds. These data are discussed later in Section 4.4. Other receptor modulators.

## 3. Diseases Associated with Altered Zonulin Levels

Most of our knowledge of the zonulin pathway is derived from research on gluten-sensitive enteropathy, also known as celiac disease [125]. Exposure to intestinal bacterial components or gliadin has been shown to lead to increased zonulin release, intestinal permeability, and consequently exacerbation of worsening clinical symptoms in patients with celiac disease [7,9,11]. Recent studies have proved that intestinal zonulin plays a crucial role in the pathomechanism of other gastrointestinal diseases [8,42]. Elevated zonulin levels, associated with impaired mucosal barrier functions, have been described in non-celiac gluten sensitivity (NCGD) [126], irritable bowel syndrome (IBS) [127,128], inflammatory bowel diseases (IBD) [34,46,129], necrotizing enterocolitis [130], neonatal gastrointestinal abnormalities [75,131], and environmental enteric dysfunction [132,133].

In addition to intestinal diseases, numerous studies have reported increased zonulin levels in various liver diseases, including non-alcoholic fatty liver disease, hepatitis, cirrhosis, and hepatocellular carcinoma [14,134,135,136]. In a recent systematic review, Ghanadi et al., analyzing the related literature, concluded that elevated levels of zonulin may lead to the release of pathogens, antigens, and toxic metals from the intestine to the liver, thereby triggering inflammatory responses and subsequent liver tissue damage [14].

A remarkable body of evidence links the zonulin-induced loss in small intestinal barrier function with diabetes mellitus, as elevated serum or fecal zonulin levels may predict the onset of the disease and shows correlation with poor glycaemic control in type 1 (T1D) [137,138,139] and type 2 (T2D) diabetes patients [69,140]. The association between zonulin levels and impaired glucose metabolism has also been described in patients with obesity [62,141,142,143,144] or insulin resistance associated with polycystic ovary syndrome (PCOS) [145,146]. Moreover, it has recently been shown that elevated levels of zonulin could be a potential predictor of complications related to pregnancy, including gestational diabetes (GDM), intrahepatic cholestasis (ICP), hypertensive disorders (HDP), and adverse perinatal outcomes [131,147,148,149,150,151,152].

Similarly, increased intestinal permeability and elevated zonulin levels have been described in patients with various forms of arthritis, including rheumatoid arthritis (RA), ankylosing spondylitis, or spondyloarthropathy [59,153,154,155]. These studies suggest that the integrity of intestinal barrier may determine the severity of systemic inflammation due to the migration of immune cells from the gut into the joints [153]. In addition, a possible role of zonulin in the pathomechanism of other disorders has been suggested, including but not limited to cardiac [68,156], pulmonary [12,13,157,158], or renal diseases [15,159,160,161].

### 3.1. Central Nervous System Diseases

Emerging literary data over the past decades have revealed the link between the presence of dysbiosis, gastrointestinal disorders, and an increased risk of diseases affecting the central nervous system. Not surprisingly, a growing body of research has demonstrated the possible role of zonulin in the pathomechanism of these MGBA diseases.

High serum zonulin levels and impaired intestinal barrier functions have been reported in pediatric patients with mental disorders, including attention deficit hyperactivity disorder (ADHD) and autism spectrum disorder (ASD) [162,163]. In addition, positive correlation has been found between zonulin levels and the severity of autism as quantified by Childhood Autism Rating Scale (CARS) scores [164,165].

Similarly, elevated zonulin release has been found in adult patients with CNS diseases, including bipolar disorder [166], schizophrenia [167], anxiety, depression [73,168], Alzheimer’s disease [169], Parkinson’s disease [170], or sclerosis multiplex [171]. Most of these studies have shown that zonulin levels are associated with disease progression.

The intact BBB plays a crucial role in the protection of central neurons by regulating the penetration of circulating antigens, immune cells, inflammatory agents, toxins, and pathogens [172]. Previously, Rahman et al. have shown that brain endothelial cells express zonulin receptors, including EGFR and PAR_2_, and that the exposure of BBB to zonulin leads to its increased permeability [173]. As we described above (Section 2.2 Biological activity of zonulin), activation of zonulin receptors leads to the disruption of actin cytoskeleton and to the dislocation of ZO-1, causing the deterioration of the TJ complex. Claudin-5 is the most enriched member of TJ proteins in the BBB, and its integrity is crucial for neuroprotection [174]. Several studies have shown that increased intestinal zonulin release and permeability are associated with high serum levels of claudin-5 in patients with neuroinflammatory or neurodegenerative disorders [73,166,167,175,176].

Recently, Miranda-Ribera et al. have demonstrated a key role of the zonulin pathway in CNS diseases using a zonulin transgenic mouse strain [18]. It has been shown that high levels of zonulin resulted in increased intestinal permeability of mice and dysbiosis as a consequence of the malabsorption-related changes of luminal content. In addition, zonulin transgenic mice were characterised by impaired BBB integrity, neuroinflammation, and behavioral alterations, which were moderately ameliorated by antibiotic treatment, causing microbiota depletion in their gut.

Stuart et al. published their results on the role of zonulin in BBB integrity in a short Letter to editor, suggesting that zonulin may regulate the pathophysiological processes in neurological diseases indirectly, through the regulation of the gut–brain axis. The author did not demonstrate a direct effect of zonulin treatment on the permeability of cerebral microvascular endothelial cells [35], which contradicts the previous findings of Rahman et al. [173]. A possible explanation could be that these studies used different recombinant zonulins, but it is more likely that the reason for the different findings is the applied dose of zonulin. Indeed, Tripathi et al. demonstrated that zonulin (which was identical to Stuart’s) at concentrations of 40–200 μg/mL increased intestinal permeability in mice, but a concentration of 20 μg/mL or lower had no effect on permeability [7]. Stuart et al. used a single zonulin treatment of 15 μg/mL, which probably was too low to affect the permeability of endothelial cells.

### 3.2. Viral Infections

Partly due to the intensified research on the SARS-CoV-2 pandemic, emerging data have recently revealed the role of the zonulin pathway in viral infections, which, in addition, often have CNS involvement. Elevated zonulin levels were demonstrated in patients with SARS-CoV-2 infection, which was associated with more severe outcomes [175,177,178,179,180,181]. These studies suggest that the prolonged presence of undigested SARS-CoV-2 viruses leads to enhanced zonulin release in the gastrointestinal tract, resulting in impaired intestinal permeability, which goes together with the accelerated trafficking of viral antigens into the bloodstream, leading to hyperinflammation (causing multisystem inflammatory syndrome—MIS). Moreover, a high serum level of zonulin leads to the disruption of BBB allowing viruses to penetrate into the brain and cause severe neurological symptoms as well [178,180].

Accordingly, similar findings have been demonstrated in connection with the human immunodeficiency virus (HIV), showing the connection between elevated serum zonulin levels and worsening gastrointestinal symptoms or decreased liver function [182,183,184].

Increased zonulin levels were also reported in patients with hepatitis B virus-associated chronic hepatitis [134]. However, decreased serum zonulin amounts were measured in patients with hepatitis B and C virus infection [185,186], which are somehow contradictory to the substantial amount of literary data on zonulin in different liver diseases [14]. Although the authors gave no explanation for decreased serum zonulin levels, the reason for the observed phenomenon could be a technical issue, which is discussed later in Section 5.2. Technical issues.

## 4. Zonulin Pathway as a Therapeutic Target

The integrity and thus the function of BBB TJs play a crucial role in the pathomechanism of neuroinflammatory and neurodegenerative diseases. Previously, it has been suggested that targeting different elements of the zonulin pathway, including actin filaments, TJs, or NF-κB, have potential therapeutic effects on CNS diseases. Indeed, encouraging results are accumulating from a recent preclinical study, using myosin light chain kinase (MLCK) inhibitor ML-7, which attenuates BBB disruption by preventing the disintegration of actin cytoskeletal microfilaments [187]. Similarly, blocking the cleavage of TJ proteins by matrix metalloproteases (MMP) inhibitors, using either direct (broad-spectrum or selective MMP-2 and MMP-9) [188,189] or indirect inhibitors (COX) [190] has been shown to protect BBB. Peroxisome proliferator-activated receptor-γ (PPAR-γ) agonists, such as rosiglitazone, pioglitazone, or D-allose, also prevented BBB integrity by inhibiting NF-κB activation [191,192,193,194]. Therefore, the use of zonulin inhibitors seems to be justified in the treatment of CNS diseases.

### 4.1. Human Studies with Larazotide Acetate

Over the past decade, larazotide acetate (also known as AT-1001), a pharmacological inhibitor of the zonulin pathway, has received increasing attention. Firstly, Wang et al. published a synthetic oligopeptide (GGVLVQPG) in 2000, representing an N’-terminal sequence of zonulin, which had a strong inhibitory effect on receptor binding of zonulin [11]. Since then, a large amount of knowledge has accumulated on this competitive zonulin inhibitor, demonstrating its strong effect on the regulation of TJs and making it one of the most promising therapeutic candidates for celiac disease [195]. Several interventional human studies have demonstrated good tolerability and beneficial effects of larazotide acetate on intestinal permeability (Table 4).

As larazotide acetate has successfully passed phase I and II clinical trials, the scientific community has raised the opportunity of its expanded access. As discussed above in the Section 3.2. Viral infections, a role for zonulin has been suggested in the pathomechanism of COVID-19-associated complications. Accordingly, short time proof-of-concept studies in a limited number of enrolled patients have shown that treatment with larazotide acetate improves the clinical manifestations of MIS-C by reducing gastrointestinal symptoms and the severity of systemic inflammation (Table 4) [178,210]. Now, its efficacy is under investigation in a phase II, randomized, double-blind, placebo-controlled clinical trial in patients with MIS-C [211]. In addition, the potential use of larazotide acetate in the treatment of metabolic diseases, including insulin resistance, diabetes mellitus, or non-alcoholic fatty liver disease (NAFLD), as well as to improve glucose and lipid metabolism of patients, has been hypothesized [212].

### 4.2. Preclinical Studies with Larazotide Acetate

Recently, human and basic research studies have revealed that high zonulin levels may affect the permeability of not only the intestine but also of other organs. Therefore, numerous preclinical studies have aimed to investigate the efficacy of larazotide acetate in experimental animal models of various diseases. Briefly, treatment with larazotide acetate has been shown to improve epithelial barrier function, thereby attenuating the severity of the investigated disorders, including colitis, vasculitis, fibrosis, arthritis, and respiratory or liver diseases (Table 5).

### 4.3. Future Perspectives of Zonulin Antagonists

Larazotide acetate was originally created as an orally administered drug with minimal absorption as the primary target cells were intestinal epithelial cells [195]. The oral administration of a therapeutic peptide can be challenging especially for compounds with expected systemic effect [226]. Systemic drugs have to penetrate the intestinal barriers, including a thick mucus gel and epithelial layer before being digested by luminal enzymes. The phase II clinical trial to evaluate the efficacy and tolerability of larazotide acetate showed that plasma levels were below the quantification limit (0.5 ng/mL) even after 7 or 14 days of daily treatment [198]. Therefore, no systemic effect should be expected after the per os treatment with larazotide acetate. At the same time, as shown in Table 5, summarizing the different methods of administration, intratracheal, intravenous, or intraperitoneal administration of larazotide acetate produced beneficial effects.

The original drug has to undergo further pharmacological development for extraintestinal use. Recent studies have reported that modification of larazotide acetate or its derivates has improved lipophilicity and intestinal absorption [227,228,229]. The resulting compound retained the biological activity of larazotide acetate and was detectable (20–30 ng/mL) in the plasma of mice after a single per os administration [229].

Besides larazotide acetate, there is another synthetic zonulin-related peptide fragment known as AT-1002, which, unlike larazotide acetate (AT-1001), has proved to be an agonist of zonulin receptors. Indeed, treatment of epithelial or endothelial cells with AT-1002 led to increased permeability by reversible opening of TJs [230,231]. Since its discovery, AT-1002 has become an important permeability-modulating component in drug development that can be used to increase the absorption and distribution of other drugs [230,232]. Several studies showed that AT-1002 can be used to increase intestinal, intranasal, intratracheal, or transdermal penetration of various compounds improving their bioavailability [233].

These preclinical data suggest that larazotide acetate or other zonulin receptor modulators (by choosing the appropriate route of administration) may prevent BBB integrity and should be investigated in CNS-related diseases, as well.

### 4.4. Other Receptor Modulators

Although binding to zonulin receptors, including PAR_2_ and EGFR, leads to the disruption of TJs, literary data on modulation of PAR_2_ and EGFR by inhibitors other than larazotide acetate are confusing (Table 6).

Recently, it has been shown that, in contrast to larazotide acetate, peptidic antagonists of PAR_2_, including FSLLRY-NH_2_ or SLIGRL-NH_2_, decreased the expression of ZO-1 and claudin-1 and destroyed the barrier function of nasal epithelial cells [121]. Similarly, a small molecule antagonist, GB83, exerted harmful effects on colon epithelial cells by decreasing the expression of autophagy- and TJ-related factors and increased permeability [234]. In contrast, inhibition of the PAR_2_ pathway by GB88 in lung epithelial cells [235] or using I-191 in arterial endothelial cells [236] moderated actin rearrangement and TJ disruption and reduced the permeability of the cellular monolayers. Moreover, a non-peptidic PAR_2_ ligand, the full agonist AC-55541, ameliorated the IL-17-induced loss of epithelial resistance in brain microvascular endothelial cells [237].

The EGFR tyrosine kinase inhibitor AG1478 also prevented TJ disassembly and epithelial resistance impairment in microvascular endothelial cells modeling BBB [238], in lung epithelial-like cells [239], and in oral epithelial tumour cells [240]. In contrast, decreased expression of TJs, barrier dysfunction, and increased permeability were induced by other EGFR tyrosine kinase inhibitors, such as erlotinib [241], gefitinib, icotinib [242], or dacomitinib [243,244] in intestinal epithelial cells. Similar effects were found in other cell types after treatment with lapatinib [245] or vandetanib [246]. These studies suggest that these compounds have a significant impact on the complex signaling pathway of EGFR, triggering stress responses, and finally leading to cell death [242]. This phenomenon may be the underlying molecular mechanism of diarrhea, which is one of the most frequent side effects of second-generation EGFR inhibitors [247].

All these data together suggest that PAR_2_ or EGFR modulators could be used to regulate epithelial or endothelial barrier function, considering that the applied drug should affect the PPI-DAG-PKC pathway, which plays a central role in zonulin-induced TJ disruption, but not ERK, JNK, or Akt signaling, which are essential for the physiological regulation of basic cellular processes, including cell growth, survival, proliferation, and apoptosis [248].

**Table 6 ijms-24-07548-t006:** Effect of PAR_2_ and EGFR modulators on TJ integrity and/or transcellular permeability of epithelial or endothelial cells based on literary data.

Target	Type	Compound	Cell Line	Effect on TJs and/or Transcellular Permeability	Ref.
PAR_2_	peptidic antagonist	FSLLRY-NH_2_	pHNECs	harmful	[121]
		SLIGRL-NH_2_			
	non-peptidic full agonist	AC-55541	hBMECs	protective	[237]
	small molecule antagonist	GB88	A549		[235]
			hECs		[236]
		GB83	Caco2	harmful	[234]
EGFR	tyrosine kinase inhibitor	AG1478	hCMEC/D3	protective	[238]
			Calu-3		[239]
			HSC-3		[240]
		erlotinib	IEC-6	harmful	[241]
		gefitinib			[242]
		icotinib			
		dacomitinib	T84		[244]
		lapatinib	HBCCs		[245]
		vandetanib	Calu-6		[246]

Abbreviations: pHNECs: primary human nasal epithelial cells; hBMECs: human brain microvascular endothelial cells; hECS: primary human arterial endothelial cells; hCMEC: primary human cardiac microvascular endothelial cells; HBBCs: primary human breast cancer epithelial cells.

## 5. Considerations

### 5.1. Nomenclature

During the preparation of the present review, several anomalies were identified in the literature on zonulin. Some of them stemmed from the incorrect use of the zonulin-related nomenclature. Indeed, at some point in writing the present manuscript, a Google Scholar search gave 74 hits using “zonulin (ZO-1” as a search term, implying that at least these studies considered the two different proteins to be identical. What is even more astonishing is that the amount of zonulin as a biomarker was investigated in many of these articles [69,90,108,176,249]. As discussed above (in Section 2.2 Biological activity of zonulin), zonulin and ZO-1 are related. Zonulin is a secreted protein, which binds to its receptors to induce cytoskeletal reorganisation and TJ disruption. ZO-1 is a member of the TJ system responsible for the cross-linking of transmembrane TJ proteins (e.g., claudin, occludin) with the actin cytoskeleton. Overall, zonulin and ZO-1 are not identical—ZO-1 is more of a target of zonulin, which explains why the expression of these proteins usually changes in opposite ways (Table 3).

Similarly, numerous studies use the appellation ‘zonulin-1’ (595 hits in Google Scholar), which is a non-existing protein, but a mixture of the denominations of zonulin and zonula occludens 1 [14,69,89]. It can be assumed that the biological interpretation of the findings from these studies is questionable, or at least confusing.

Similarly, the synonymy of larazotide acetate may be a source of confusion, as other compounds are also known as AT-1001. Indeed, migalastat hydrochloride (Galafold), a pharmacological chaperone drug approved by FDA for Fabry disease is also referred to as AT-1001 in clinical trials [250,251]. Moreover, AT-1001 is also a synonym for an α3β4 nicotinic acetylcholine receptor antagonist, a potential therapeutic agent for smoking cessation [252].

### 5.2. Technical Issues: Zonulin as a Biomarker and Therapeutic Target

Much of our knowledge on zonulin is due to the work of Fasano and his colleagues, including the effect of Zot derived from *Vibrio cholerae* on intestinal permeability, the discovery of its human analogue zonulin, the identification as pre-haptoglobin 2, the description of the regulation of zonulin release, the exploration of the underlying molecular mechanisms, and biological outcomes of zonulin receptor activation. These studies and their findings are reasonably coherent and follow a logical path; however, some results may not be sufficiently supported by independent experiments. This is perhaps one possible reason for the increasing number of controversies about zonulin in recent years.

Over the past decade, several studies have examined serum zonulin levels in various diseases, and along with that, many of these publications have demonstrated that zonulin cannot be used as a biomarker of increased intestinal permeability [253,254,255,256]. Emerging evidence has revealed that the controversial results of these studies may be due to some commercially available enzyme-linked immunosorbent assays (ELISAs) to specifically detect zonulin [93,257,258,259,260].

These issues led to a series of short PostScript publications on the pages of Gut, a prestigious journal of gastroenterology. Massier et al. explained that the controversial results of commercial ELISAs may be due the fact that the first published sequence of zonulin, against which the ELISAs were developed, does not cover all different zonulin sequences [261]. This is supported by the fact that the sequence of proteins isolated with anti-Zot antibodies and identified as zonulin by Fasano et al. [7,11] does not contain the octapeptide sequence of larazotide acetate, which was initially used as a zonulin receptor antagonist [262]. The peptide fragment in question is rather an immunoglobulin sequence, which can be easily confirmed by a protein query using NCBI Protein BLAST. In addition, Massier et al. point out that the measurement of zonulin levels in preclinical studies is highly questionable, as pre-haptoglobin-2 is a human-specific protein and is not naturally expressed in rodents [261].

In his reply, Fasano provided some clarifications [263]. Briefly, the author pointed out that zonulin is rather a family of structurally and functionally related proteins (zonulin family peptides—ZFPs), including not only pre-haptoglobin-2 but also other mannose-binding lectin-associated serine proteases (MASPs), such as properdin, coagulation factor X, or CD5 antigen [264]. Indeed, in recent years, an increasing number of studies have introduced the expression of ZFP or zonulin-related protein (ZRP) instead of zonulin or pre-haptoglobin-2 [144,257,259,265]. Fasano concluded that despite the possible non-specificity of commercial ELISAs, which should be clarified to improve their reliability, the overall impact of the zonulin pathway on diseases associated with altered tissue permeability is unequivocal. This perspective is consistent with the fact that zonulin or related proteins (ZRPs, ZFPs) have been detected in human patients with haptoglobin 1-1 homozygous genotype [35] and in various rodents (Table 3), which have been shown not to express pre-haptoglobin-2. Moreover, treatment with zonulin receptor antagonist larazotide acetate has also shown a protective effect in numerous preclinical studies in non-humanized mice or rats as well (Table 5).

Choosing the appropriate zonulin ELISA kit is a challenge. The most commonly used kit, manufactured by Cusabio, was developed to detect pre-haptoglobin-2, but its cross-reactivity with properdin, pre-haptoglobin-1, or mature haptoglobins was not investigated. However, the exact protein sequence of the immunogen epitope and that of the standard protein provided to the product is not public. The specificity of Elabscience’s is also questionable as both the full-length recombinant pre-haptoglobin-2, a part of mature haptoglobin sequence, and *Vibrio cholerae*-derived Zot was assigned as the target of the applied antibody. Immundiagnostik offers an ELISA intended for the determination of ZFPs, based on a polyclonal antibody against zonulin sequence published by Wang [11], which, as discussed above, was isolated by anti-Zot antibodies and does not overlap with the known sequence of pre-haptoglobin-2. Clarifying these issues is essential for the proper interpretation of measured data, and in addition, may clarify the possible role of other ZFP members in the pathomechanism of various diseases.

## 6. Summary

In the present article, we summarized recent literary data on the potential role of zonulin and its receptors in the MGBA-related diseases (Figure 1).

Luminal agents, including gliadin and microbiota components, may induce zonulin release from the intestinal epithelial cells through CXCR3 receptor activation. It has been shown that while Gram-negative bacteria mostly facilitate this process, some bacterial strains, including *Bifidobacterium* and *Lactobacillus* spp. decrease intestinal zonulin production.

Free zonulin binds to its cell surface receptors, EGFR and PAR_2_, leading to cytoskeletal rearrangement and TJ-disassembly of epithelial cells causing increased intestinal permeability. Consequently, the impaired intestinal barrier facilitates the penetration of luminal agents and promotes intestinal inflammation. The immunogenic substances and proinflammatory cytokines may also enter the bloodstream, affecting BBB and other tissues. Brain endothelial cells also express zonulin receptors, thereby the circulating zonulin can directly increase the permeability of BBB, facilitating neuroinflammation. This process may explain the observation that CNS diseases are often associated with dysbiosis and increased serum zonulin levels, which correlate with the deterioration of cognitive functions.

Several promising studies have shown that intestinal permeability can be normalized pharmacologically by modulating zonulin receptors, such as larazotide acetate. In addition to preserving the function of the intestinal barrier and thereby reducing the levels of proinflammatory factors in the blood (which may have a neuroprotective effect in itself), zonulin-antagonists, PAR_2_ modulators, or EGFR inhibitors may be useful tools to reduce neuroinflammation by acting directly on the endothelial cells of BBB.

Although there are pharmaceutical challenges and technical issues to be solved, our knowledge of zonulin suggests that it may play a crucial role both in intestinal and CNS diseases and may serve as a potential therapeutic target.

## Figures and Tables

**Figure 1 ijms-24-07548-f001:**
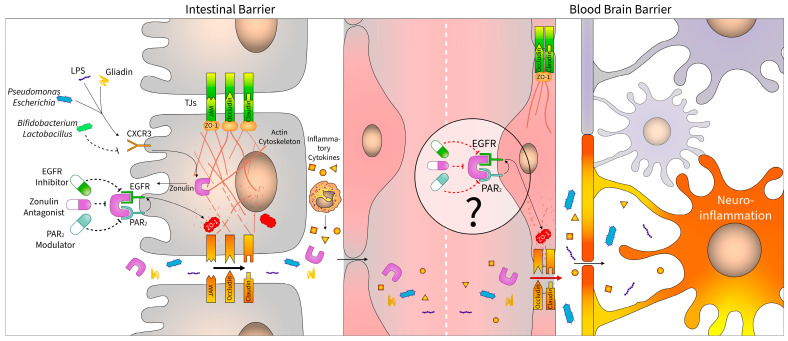
Zonulin as a potential therapeutic target in the disorders of the central nervous system. Luminal components, including gliadin and bacteria, induce zonulin production of the intestinal epithelial cells via CXCR3. Zonulin is a TJ modulator protein, which activates EGFR and PAR_2_ receptors and thereby induces actin and ZO-1 disassembly, leading to increased paracellular permeability through TJ disruption. Immunogen fragments and proinflammatory cytokines enriched in the bloodstream induce neuroinflammation, which is further facilitated by circulatory zonulin released from the intestine. Inhibition of the zonulin pathway by using zonulin antagonists (e.g., larazotide acetate), EGFR inhibitors, or PAR_2_ modulators can preserve the barrier function of epithelial and endothelial layers in the intestine and, presumably, also in the brain. Abbreviations: CXCR3: C-X-C chemokine receptor type 3; EGFR: epidermal growth factor receptor; JAM: junctional adhesion molecule; LPS: bacterial lipopolysaccharide; PAR_2_: proteinase activated receptor 2; TJ: tight junction; ZO-1: zonula occludens 1.

**Table 1 ijms-24-07548-t001:** Correlation between zonulin levels and the abundance of certain bacterial species based on literary data of descriptive human studies.

Gut Microbiome Member	Population	Zonulin Levels in Relation with Microbial Abundance	Ref.
*Escherichia coli*	ankylosing spondylitis patients	↑	[59]
	relatively healthy elderly volunteers	↑	[60]
	healthy adult volunteers	↑	[61]
*Bacteroides*	normal weight and obese volunteers	↑	[62]
	Hashimoto-thyroiditis patients	↑	[63]
*Prevotella*	ankylosing spondylitis patients	↑	[59]
	obese colorectal carcinoma patients	↑	[64]
*Pseudomonas*	relatively healthy elderly volunteers	↑	[60]
*Shigella*		↑	
*γ-Proteobacteria*		↑	
*Rhizobiales*		↑	
*Firmicutes*	normal weight and obese volunteers	↑	[62]
*Erysipelotrichales*	healthy women	↑	[65]
*Actinobacteria*	relatively healthy elderly volunteers	↑	[60]
*Clostridium*	healthy adult volunteers	↑	[61]
*Enteroviridae*	celiac disease with or without T1D	↑	[66]
LPS (in serum)	community-acquired pneumonia patients	↑	[67]
	precocious acute myocardial infarction patients	↑	[68]
	T1D	↑	[69]
	Graves’ disease patients	↑	[70]
	children with IgE mediated and non-IgE-mediated food allergy	↑	[71]
	vitiligo patients	↑	[72]
	adolescents with major depressive disorder	↑	[73]
	septic patients	↑	[74]
*Lachnoclostridium*	healthy newborns	↑	[75]
*Ruminococcus gnavus*		↑	
*Ruminococcus torques*		↑	
*Erysipelotrichales*		↑	
*Coriobacteriales*		↑	
*Alphaproteobacteria*		↓	
*Corynebacterium*		↓	
*Pdeudomonadales*		↓	
*Moraxellaceae*		↓	
*Staphylococcus*		↓	
*Bifidobacterium*	Hashimoto-thyroiditis patients	↓	[63]
*Lactobacillus* spp.	healthy adult volunteers	↓	[61]
*Ruminococcaceae*	healthy women	↓	[65]
*Faecalibacterium*		↓	
*Odoribacter*		↓	
*Rikenellaceae*		↓	

Abbreviations: LPS: lipopolysaccharide; Ref.: reference; T1D: type 1 diabetes; ↑: increased expression; ↓: decreased expression.

**Table 2 ijms-24-07548-t002:** Effect of mixtures of various bacterial species on zonulin levels based on literary data of randomized, interventional human clinical studies. Species separated by dashed lines indicate the elements of a multi-component treatment.

Species	Strain	Treatment and Population	Findings		Ref.
			Blood Zonulin	Fecal Zonulin	
*Lactobacillus plantarum*	CGMCC no.1258	pre- and postoperative probiotic treatment of patients operated on for colorectal carcinoma	↓	NE	[76]
*Lactobacillus acidophilus*	11				
*Bifidobacterium longum*	88				
*Lactobacillus plantarum*	CGMCC no.1258	pre- and postoperative probiotic treatment of patients operated on for colorectal carcinoma and liver metastasis	↓	NE	[77]
*Lactobacillus acidophilus*	11				
*Bifidobacterium longum*	88				
*Bifidobacterium animalis*	lactis 420	probiotic and fiber treatment of healthy overweight volunteers	↓	NE	[78]
SCM-III synbiotic mixture:		synbiotic treatment of healthy stressed individuals	↓	↓	[79]
*Lactobacillus acidophilus*	145				
*Lactobacillus helveticus*	ATC15009				
*Bifidobacterium*	420	probiotic treatment of healthy stressed individuals	-	↓	
P3T/J probiotic mixture:					
*Bifidobacterium animalis*	lactis Bi1				
*Bifidobacterium breve*	Bbr8	synbiotic and probiotic treatment of healthy stressed individuals	↓	↓	
*Lactobacillus acidophilus*	LA1				
*Lactobacillus paracasei*	101/37				
*Bifidobacterium lactis*	W51	dietary changes and probiotic treatment in obese patients	NE	↓	[80]
	W52				
*Lactobacillus acidophilus*	W22				
*Lactobacillus paracasei*	W20				
*Lactobacillus plantarum*	W21				
*Lactobacillus salivarius*	W24				
*Lactococcus lactis*	W19				
*Bifidobacterium bifidum*	W23	impact of exercise in trained men treated with probiotics	NE	↓	[81]
*Bifidobacterium lactis*	W51				
*Enterococcus faecium*	W54				
*Lactobacillus acidophilus*	W22				
*Lactobacillus brevis*	W63				
*Lactococcus lactis*	W58				
*Bifidobacterium bifidum*	W23	synbiotic treatment of healthy volunteers	-	NE	[82]
*Bifidobacterium lactis*	W51				
	W52				
*Lactobacillus acidophilus*	W22				
*Lactobacillus casei*	W56				
*Lactobacillus paracasei*	W20				
*Lactobacillus plantarum*	W62				
*Lactobacillus salivarius*	W24				
*Lactococcus lactis*	W19				
*Bifidobacterium lactis*		synbiotic treatment of children with NAFLD	-	NE	[83]
*Lactobacillus acidophilus*					
*Lactobacillus casei*					
*Bifidobacterium bifidum*	W23	probiotic treatment of migraine patients	-	-	[84]
*Bifidobacterium lactis*	W52				
*Lactobacillus acidophilus*	W37				
*Lactobacillus brevis*	W63				
*Lactobacillus casei*	W56	probiotic treatment of ulcerative colitis patients	↓	-	[85]
*Lactobacillus salivarius*	W24				
*Lactococcus lactis*	W19				
	W58				
*Bacillus subtilis*	DE111	probiotic treatment of professional baseball players	-	NE	[86]

Abbreviations: NAFLD: non-alcoholic fatty liver disease; NE: not examined; Ref.: reference; ↓: decreased level; -: no effect.

**Table 3 ijms-24-07548-t003:** Effect of various bacterial species on zonulin and/or ZO-1 levels based on literary data of experimental studies on cell lines and animal models. Species separated by dashed lines indicate the elements of a multi-component treatment.

Species	Strain	Cell Line/ Experimental Model	Findings		Ref.
			Zonulin	ZO-1	
*Escherichia coli*	6-1	CaCo2	↑	↓, disruption	[42]
		rat, rabbit, and monkey small intestinal organoids	↑	NE	
	K-12 DH5α	rabbit and monkey small intestinal organoids	↑	NE	
	21-1	rabbit small intestinal organoids	↑	NE	
	K88	4-day-old piglets	↑	↓	[92]
	K88	IPEC-J2	-	↓	
	RY13	HT-29	-	NE	[93]
	K12 DH5α		-	NE	
	042, JM221	T84	NE	disruption	[94]
	055:B5 (LPS)	CaCo2	↑	↓	[52]
		CaCo2	↑	NE	[59]
	HB101	T84	NE	disruption	[95]
*Bacteroidales* and *Escherichia coli*		malnourished mice	↑	↓	[96]
*Salmonella typhimurium*	SO1344	rabbit small intestinal organoids	↑	NE	[42]
*Pseudomonas fluorescens*		CaCo2	↑	disruption	[97]
*Prevotella*		CaCo2	↑	NE	[59]
*Acetobacter ghanensis*		CaCo2 treated with PT-gliadin	↓	-	[98]
*Porphyromonas gingivalis*		healthy mice	NE	↓	[99]
*Pseudomonas aeruginosa*		pneumonia induced in mice	NE	↑	[100]
*Fusobacterium nucleatum*		CaCo2	NE	↓	[101]
		DSS-induced colitis in mice	NE	↓, disruption	
*Ruminococcus blautia gnavus*	VPI C7-9	germ-free mice	-	NE	[102]
	CC55_001C		-	NE	
	S107-48		↑	NE	
	S47-18		-	NE	
*Clostridium difficile* toxin A and B		T84	NE	disruption	[103]
*Faecalibacterium prausnitzii* MAM		NCM460 transfection	NE	↑	[104]
		Caco2 transfection	NE	↑	
		HT-29 transfection	NE	↑	
		diabetes mellitus induced in mice	NE	↑	
*Lactobacillus rhamnosus*	GG	CaCo2 treated with gliadin	↓	↑	[91]
		HT-29	↑	NE	[93]
	P1	HT-29 treated with PT-gliadin	↓	↑	[105]
	P2		↓	↑	
	F1		↓	↑	
	P3		↓	-	
	GG		↓	↑	
*Lactobacillus casei*	C1	HT-29 treated with PT-gliadin	↓	↑	
*Bifidobacterium longum*	CECT-7347	HT-29 treated with TNF-α	NE	↑	[90]
		T84	NE	↑	[106]
		HT-29	-	NE	[93]
*Bifidobacterium* *(not specified)*		CaCo2 treated with LPS	↓	↑	[52]
		LPS-induced NEC in rats	↓	↑	
*VSL#3*		IEC-6 treated with hydrolyzed gliadin	↓	NE	[89]
		mouse small intestinal organoid treated with hydrolyzed gliadin	↓	NE	
*Lactobacillus paracasei*	D3-5	high-fat diet in old mice	NE	↑	[107]
*Lactobacillus rhamnosus*	D4-4				
	D7-5				
*Lactobacillus plantarum*	D6-2				
	D13-4				
*Enterococcus rafnosus*	D24-1				
*Enterococcus INBio*	D24-2				
*Enterococcus Avium*	D25-1				
	D25-2				
	D26-1				
*Lactobacillus paracasei*	101/37 LMG P-17504	CaCo2 treated with PT-gliadin	NE	↑	[108]
*Lactobacillus plantarum*	14 D CECT 4528				
*Bifidobacterium animalis*	lactis Bi1 LMG P-17502				
*Bifidobacterium breve*	Bbr8 LMG P-17501				
	BL10 LMG P-17500				

Abbreviations: DSS: dextran sulphate sodium; LPS: lipopolysaccharide; MAM: microbial anti-inflammatory molecule; NE: not examined; NEC: necrotizing enterocolitis; PT: pepsin/trypsin digested; Ref.: reference; VSL#3: *Streptococcus thermophilus*, *Lactobacillus plantarum*, *L. acidophilus*, *L. casei*, *L. delbrueckii* spp. bulgaricus, *Bifidobacterium breve*, *B. longum*, *B. infantis*; ↑: increased expression; ↓: decreased expression; -: no effect.

**Table 4 ijms-24-07548-t004:** Human clinical studies investigating the therapeutic applicability of larazotide acetate.

Condition	Results	Study (Enrollment)	Clinical Trials Identifier	Ref.
Healthy	good tolerability	Phase I (24)	NCT00386490	[196]
Celiac disease, gluten-free diet	good tolerability	Phase Ib (21)	NCT00386165	[197]
Celiac disease, gluten challenge	improvement in GI symptoms, good tolerability	Phase IIa (80)	NCT00362856	[198,199,200,201]
Celiac disease, gluten challenge	improvement in histological scores, good tolerability	Phase IIb (105)	NCT00620451	[202,203]
Celiac disease, gluten challenge	improvement in GI symptoms, decreased level of anti-tTG IgA	Phase IIb (171)	NCT00492960	[204,205]
Celiac disease, persistent symptoms with gluten-free diet	improvement in GI and extra-GI symptoms, good tolerability	Phase IIb (342)	NCT01396213	[206,207]
Celiac disease, gluten-free diet	(terminated based on interim analysis)	Phase III (307)	NCT03569007	[208,209]
COVID19—MIS-C	improvement in clinical symptoms, decreased level of inflammatory markers and SARS-CoV-2 nucleocapsid (N) protein	case report (1)		[178]
COVID19—MIS-C	improvement in GI symptoms, decreased level of SARS-CoV-2 Spike (S) protein	case series (4)		[210]
COVID19—MIS-C	(not completed)	Phase IIa (20)	NCT05022303	[211]

Abbreviations: Ref.: reference; GI: gastrointestinal; MIS-C: Multisystem Inflammatory Syndrome in Children.

**Table 5 ijms-24-07548-t005:** Preclinical animal studies investigating the therapeutic applicability of larazotide acetate.

Model	Species	Administration	Daily Dose	Results	Ref.
celiac disease	gliadin-sensitized HLA-HCD4/DQ8 transgenic mouse	p.o. gavage	0.25 mg	reduced intestinal permeability and macrophage infiltration	[213]
		p.o. gavage	0.3 mg	reduced intestinal permeability	[214]
intestinal permeability	*Il10*^−/−^ mouse	p.o. gavage	5 mg	reduced intestinal permeability and inflammation	[215]
spontaneous colitis		p.o. in drinking water	0.1 or 1 mg/mL	reduced intestinal permeability and inflammation	[216]
DSS induced colitis	zonulin transgenic mouse	p.o. in drinking water	1 mg/mL	reduced intestinal permeability	[217]
radiation-induced enteropathy	mouse	i.p.	0.25 mg	improved clinical state and histological scores, inhibited bacterial translocation, elevated TJ protein levels	[218]
healthy (pharmacokinetics)	pig	p.o. capsule	0.05 mg/kg	determining pharmacokinetics of larazotide acetate in the small intestine	[219]
*Ruminococcus blautia gnavus* colonization	germ-free mouse	p.o. in drinking water	0.15 mg/mL	reduced intestinal permeability	[102]
spontaneous T1D	BB diabetic-prone rat	p.o. in drinking water	0.01 mg/mL	inhibited development of diabetes	[138]
rheumatoid arthritis	mouse	p.o. in drinking water	0.15 mg/mL	attenuated arthritis	[153]
	*Il10ra*^−/−^ mouse, *Cldn8*^−/−^ mouse	p.o. gavage	2 × 0.05 mg	reduced intestinal permeability, inflammation, and joint swelling	[220]
vasculitis	mouse	i.p.	0.5 mg	reduced intestinal permeability and LPS translocation, prevented cardiovascular lesions	[221]
LPS-induced acute lung injury		i.t.	0.05 mg	reduced severity, decreased inflammatory markers	[12]
		i.v.	0.01 or 0.025 or 0.05 mg		
influenza		i.v.	0.15 mg	reduced severity of acute lung injury	[222]
salivary gland fibrosis		i.p.	5 mg/kg	improved epithelial barrier function, ameliorated fibrosis	[223]
NAFLD		p.o. in drinking water	0.1 or 1 mg/mL	reduced intestinal permeability	[224]
		p.o. gavage	2 × 0.03 or 2 × 0.3 mg		
acute liver failure	rat	p.o. in drinking water	0.01 mg/mL	decreased intestinal damage	[225]
		p.o. gavage	2 × 0.03 mg		

Abbreviations: Ref.: reference; p.o.: per os; i.p.: intraperitoneal; i.v. intravenous; DSS: dextran sulphate sodium; TJ: tight junction; T1D: type 1 diabetes; LPS: lipopolysaccharide; NAFLD: non-alcoholic fatty liver disease.

## Data Availability

Not applicable.

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
