# Peer review of "Zonulin as a Potential Therapeutic Target in Microbiota-Gut-Brain Axis Disorders: Encouraging Results and Emerging Questions"

_ijms, 2023, doi:10.3390/ijms24087548_

Round 1

Reviewer 1 Report

The paper looks good. Title should be simple something like “Zonulin: A Potential Therapeutic Approach for Microbiota-Gut-Brain Communication”

Other than this authors needs to do some of these changes

1.    in abstract, Instead of "during the last decade", you could specify the timeframe, for example "in the past 10 years". In sentence 10, consider rephrasing to make it clearer that microbial alterations cause increased intestinal permeability, which then leads to inflammation and affects distant organs.In sentence 12, consider specifying the importance of the intestinal epithelial barrier in the microbiota-gut-brain axis. In sentence 13, consider adding that zonulin regulates tight junctions between cells in the intestinal epithelium. In sentence 14, consider specifying how zonulin regulates blood-brain barrier function.In sentence 15, consider specifying which specific pharmaceutical approaches have been studied to modulate zonulin-associated pathways. In sentence 16, consider adding which specific emerging issues related to zonulin are discussed in the review.

2.    Introduction, Consider starting with a more attention-grabbing opening sentence to engage the reader and introduce the topic in an intriguing way. Define MGBA and explain its significance more clearly. Provide more context about the prevalence and impact of neurological disorders and the need for research in this area. Simplify some of the language to make it more accessible for readers who may not be familiar with scientific terminology. Add more detail about the scope and focus of the review, including what specific questions or areas of research it will address.

3.    In section, 2.2. Regulation of zonulin, authors are advised to do these changes. Add notes on these statements. Further research can be done to elucidate the underlying mechanism of microbiota's impact on zonulin release. Specifically, the effects of different bacterial strains on zonulin levels can be explored to identify the specific strains that induce or inhibit zonulin release. The potential clinical implications of zonulin dysregulation can be investigated. For instance, the passage mentions that gluten exposure can lead to zonulin release and harmful effects on intestinal epithelial cell viability and permeability, even in non-coeliac patients. Therefore, understanding the role of zonulin dysregulation in different disease states can help in developing targeted therapies. Since the passage suggests that some bacterial strains can cleave gluten peptides and inhibit gliadin-induced cytotoxic responses in intestinal epithelial cells, further research can be done to explore the use of probiotics as a potential therapeutic option in diseases that involve zonulin dysregulation. Lastly, the passage mentions that zonulin release can lead to intestinal barrier damage and dysfunction, which can facilitate the invasion of parasites into the lamina propria. Therefore, exploring the role of zonulin in parasitic infections can be an interesting area of research.

4.    5.2. Technical issues: zonulin as biomarker and therapeutic target: it seems that there are some controversies and discrepancies surrounding zonulin, its detection, and its role in diseases related to altered tissue permeability. Here are some suggestions for further exploration and consideration to give some writings: Verify the specificity and reliability of commercial enzyme-linked immunosorbent assays (ELISAs) used to detect zonulin. It appears that some ELISAs may not be suitable for specific detection of zonulin due to incomplete coverage of zonulin sequences. This can lead to conflicting results and inaccurate interpretation of zonulin levels in various diseases. Explore the possibility that zonulin is a family of structurally and functionally related proteins (zonulin family peptides – ZFPs) that includes not only pre-haptoglobin-2 but also other mannose-binding lectin-associated serine proteases (MASPs). This may provide a more comprehensive understanding of the molecular mechanisms and biological outcomes of zonulin receptor activation. Consider using the expression of ZFP or zonulin related protein (ZRP) instead of zonulin or pre-haptoglobin-2 in studies to avoid possible non-specificity of commercial ELISAs. This may improve the reliability of zonulin detection and interpretation of its role in various diseases. Investigate the impact of zonulin-pathway on diseases related to altered tissue permeability using different animal models and human patients with haptoglobin 1-1 homozygous genotype. This may help to determine the overall impact of zonulin and related proteins (ZRPs, ZFPs) on these diseases and provide more accurate and reliable data for clinical applications. Evaluate the effectiveness of treatment with zonulin receptor antagonist larazotide acetate in preclinical studies performed on non-humanized mice or rats. This may provide insights into potential therapeutic strategies for diseases related to altered tissue permeability.

Author Response

Dear Reviewer, 

Thank you for your effort to review our manuscript and also for your valuable suggestions and comments.

Based on your and other reviewers criticism, we have modified the title of our manuscript. As stated in the 'Instruction for Authors', the second half of the title indicates the content and the type of the study.

According to your suggestions, the abstract was revised.

Based on the GoogleScholar search results, the 'last decade' was specified in 10 years. The emerging number of MGBA-related articles can be seen in figure below. 

Introduction was also revised and completed with paragraphs about MGBA and also with the aim of the review.

Section 2.2 Regulation of zonulin was completed at several points in accordance with your suggestions about the possible importance of zonulin in other diseases, the opportunity of therapeutic usage of beneficial microbiota, and the role of zonulin in parasitic infections.

A new paragraph can be found in section 5.2. Technical issues, where - based on your suggestions - we detailed the specificity of commercial zonulin ELISAs and noted, that resolving those contradictions may help not only to clarify the role of zonulin (preHp2) as biomarker but also to reveal the impact of other ZFP members in several deseases.

Reviewer 2 Report

The article is good and can be accepted after minor edits.

Author Response

Dear Reviewer, 

Thank you for reviewing our article and the positive assessment of our study.

Reviewer 3 Report

In the present manuscript, the authors have discussed the microbiota-gut-brain axis with special emphasis on intestinal zonulin. The authors have gathered important literature and discussed comprehensively the mechanisms and potential therapeutic targets to modulate the zonulin-associated pathways. Overall the manuscript is well written and I recommend it for acceptance with minor spell checks.

Author Response

Dear Reviewer,

Thank you for your effort to review our manuscript.

Reviewer 4 Report

This review focused on the function of Zonulin. It aims at its role in Microbiota-Gut-Brain Axis Signaling yet the summarization is not comprehensive enough.

For example, the section 3.1 immune-mediated disease is lack of the detailed knowledge and findings on the immunology

In addition, the review of zonulin in Microbiota-Gut-Brain Axis Signaling is limited.

Author Response

Dear Reviewer,

Thank you for your effort to review our article.

We agreed with your criticism, using the term 'signaling' may suggest that molecular pathways or signal transduction will be descussed, so we modified the title of our manuscript and clarified the aim of the review.

In section 3.1 we aimed to summarize diseases associated with increased zonulin levels and which affect various organs but not the CNS, instead of discuss the direct immunoregulatory role of zonulin. As the majority of these disorders are immune-mediated, we found this as a most appropriate definition. To avoid the missunderstanding, based on your comment we have made changes in the section and subsection titles and also in the main text.

Reviewer 5 Report

I would like to thank the Authors of this manuscript, as well as the Editors of the journal, to allow me the opportunity to provide comments on it.

It is my understanding that an extensive literature review has been carried out to define the role of the protein zonulin as a therapeutic target in microbiota-gut-brain axis signaling.

The review is overall very well constructed and properly organized, with a clear objective and several interesting paragraphs analysing what is known around zonulin in terms of physiology and pathological responses. I find the link with intestinal dysbiosis and blood-brain barrier permeability very fascinating, especially in the context of behavioral, psychiatric and degenerative disorders. My only objections would be that there is a bit of redundancy between the introductory paragraph and the first section, so those may be slightly trimmed so as not to repeat the same information twice unnecessarily, and that maybe a Figure could be put earlier in the manuscript, instead of just having the one at the end of it.

Author Response

Dear Reviewer, 

Thank you for your efforts ro review our article and also for your comments.

Introduction was reviesed and completed with informations about the MGBA, which are discussed only in this section. We agree, that the figure could be placed earlier, helping the understand of the molecular mechanism of zonulin pathway. However, the figure illustrates the statement of the summary, and each of the elements can be placed in context after reading the majority of the main text. It occured to us that the figure could serve as graphical abstract, but we found it too complex compared to what is expected in IfA. Thereby, we left the figure at the Summary, while referring to it in earlier sections.

Round 2

Reviewer 4 Report

Thank you for trying to address my comments. However, the summary of Microbiota-Gut-Brain Axis is still weak.

Author Response

Thank you again for taking time to assess our manuscript. 

Best regards